# Crystal structures of the CPAP/STIL complex reveal its role in centriole assembly and human microcephaly

Matthew A Cottee[1†], Nadine Muschalik[1†], Yao Liang Wong[2], Christopher M Johnson[3], Steven Johnson[1], Antonina Andreeva[3], Karen Oegema[2], Susan M Lea[1], Jordan W Raff[1], Mark van Breugel[3]*

[1]Sir William Dunn School of Pathology, University of Oxford, Oxford, United Kingdom; [2]Department of Cellular and Molecular Medicine, Ludwig Institute for Cancer Research, University of California, San Diego, La Jolla, United States; [3]Laboratory of Molecular Biology, Medical Research Council, Cambridge, United Kingdom

**Abstract** Centrioles organise centrosomes and template cilia and flagella. Several centriole and centrosome proteins have been linked to microcephaly (MCPH), a neuro-developmental disease associated with small brain size. CPAP (MCPH6) and STIL (MCPH7) are required for centriole assembly, but it is unclear how mutations in them lead to microcephaly. We show that the TCP domain of CPAP constitutes a novel proline recognition domain that forms a 1:1 complex with a short, highly conserved target motif in STIL. Crystal structures of this complex reveal an unusual, all-β structure adopted by the TCP domain and explain how a microcephaly mutation in CPAP compromises complex formation. Through point mutations, we demonstrate that complex formation is essential for centriole duplication in vivo. Our studies provide the first structural insight into how the malfunction of centriole proteins results in human disease and also reveal that the CPAP–STIL interaction constitutes a conserved key step in centriole biogenesis.

**\*For correspondence:**
vanbreug@mrc-lmb.cam.ac.uk

[†]These authors contributed equally to this work

**Competing interests:** The authors declare that no competing interests exist.

**Reviewing editor**: John Kuriyan, Howard Hughes Medical Institute, University of California, Berkeley, United States

## Introduction

Centrioles are small cylindrical organelles whose outer walls contain a ninefold symmetric array of microtubule triplets. These structures form the basal bodies that template the assembly of cilia and flagella, and they also organise a proteinaceous matrix termed the pericentriolar material (PCM) to form centrosomes, the main microtubule organising centres in animal cells. These organelles play an important part in many aspects of cell organisation, and centriolar dysfunction is linked to a plethora of human diseases, including cancer, obesity, macular degeneration and polycystic kidney disease (*Nigg and Raff, 2009*; *Bettencourt-Dias et al., 2011*).

Recently, an unexpected genetic link has emerged between centriole/centrosome assembly and human brain size. Autosomal recessive primary microcephaly (MCPH) is a rare condition where patients are born with small brains (*Thornton and Woods, 2009*). All eight identified MCPH genes encode proteins that localise to centrioles and/or centrosomes/spindle poles (*Thornton and Woods, 2009*; *Hussain et al., 2012*). It is unclear why mutations in these proteins are linked to such a specific neuro-developmental problem in humans, but it seems likely that some aspect of centriole/centrosome function must be particularly important for the proper proliferation of human neural progenitors (*Siller and Doe, 2009*; *Megraw et al., 2011*). In support of this possibility, mutations in the centriolar components CPAP (DSas-4 in *Drosophila*, here called dCPAP) and STIL (Ana2 in *Drosophila*, here called dSTIL) in flies lead to defects in the asymmetric division of larval neural stem/progenitor cells (*Basto et al., 2006*). Mutations in MCPH proteins in mice, however, lead to complex phenotypes that can include,

**eLife digest** Organisms—and individual tissues—grow and develop by dividing their cells. However, the process of cell division does not have to be symmetric, and the fates of the cells can be very different if cellular contents, including RNAs or proteins, are exclusively retained in the 'mother' or passed to her 'daughter'. Organelles known as centrioles can play an important part in influencing whether cell division is symmetric or asymmetric.

Centrioles contain ordered assemblies of various proteins, and mutations in some of these proteins can cause developmental defects in humans. For example, mutations in the centriolar proteins CPAP and STIL cause a syndrome known as microcephaly, in which the brain is smaller than normal. Although CPAP and STIL are known to bind each other, how they interact on a molecular level to form centrioles—and how this interaction is disrupted in microcephaly—is not well understood.

Cottee et al. have now used structural and biochemical assays to explore how these two proteins bind to each other, and have identified specific amino acid residues that enable this interaction. These residues are highly conserved across many organisms, and a mutation in one of them has previously been associated with microcephaly in humans. Now, Cottee et al. demonstrate that this mutation weakens the interaction between CPAP and STIL in vitro.

To explore these processes in vivo, Cottee et al. studied mutant fruit flies in which the interactions between CPAP and STIL were weaker than normal, and found that these mutations prevented the normal formation of centrioles. Furthermore, there was a striking correlation between the ability to form centrioles in fruit flies and the ability of CPAP and STIL to bind each other, based on the structural model and in vitro binding studies.

Cumulatively, these findings reinforce the importance of CPAP and STIL in centriole formation, and suggest that one reason for the development of microcephaly may be defects in the proper formation of centrioles.

but are not restricted to, microcephaly (*McIntyre et al., 2012*). Moreover, compelling genetic links are now emerging between centrioles/centrosomes and DNA damage repair (DDR) pathways: mutations in certain MCPH genes and in genes encoding other centriole/centrosome proteins can lead to Seckel syndrome and MOPD, pathologies normally associated with defects in DDR (*Megraw et al., 2011*). Thus, the cellular mechanisms that lead to pathology when centriole/centrosome proteins are mutated in humans remain unclear.

Centrioles are complex structures, but work in several model systems revealed only a small number of conserved proteins to be important for centriole assembly. These include PLK4/SAK, SAS-6, STIL/Ana2, CPAP/CenpJ/SAS-4, Cep152/Asl, and CEP135 (*Brito et al., 2012*; *Gonczy, 2012*). Several studies have identified a complex web of putative interactions between these proteins (*Cizmecioglu et al., 2010*; *Dzhindzhev et al., 2010*; *Hatch et al., 2010*; *Tang et al., 2011*; *Vulprecht et al., 2012*; *Lin et al., 2013*). However, an understanding of centriole architecture and its assembly mechanisms will ultimately require high-resolution structures of the key centriolar components and their complexes. The power of combining structural studies with protein biochemistry and functional in vivo experiments has been demonstrated by work on SAS-6. These studies revealed how SAS-6 homo-oligomerises to organise the central cartwheel (*Kitagawa et al., 2011b*; *van Breugel et al., 2011*), the earliest structurally defined intermediate in centriole assembly (*Brito et al., 2012*; *Gonczy, 2012*), and suggested how SAS-6 might interact with SAS-5, the proposed STIL homologue in worms (*Qiao et al., 2012*). Additionally, high-resolution structures of Sak/Plk4 fragments have recently been solved (*Leung et al., 2002*; *Slevin et al., 2012*). However, equivalent studies with other core centriolar components or especially their complexes are currently missing, and how any of these proteins might be structurally and mechanistically compromised in MCPH is not known.

Of particular interest in this regard is the putative centriolar CPAP–STIL complex, as mutations in both components result in MCPH (*Leal et al., 2003*; *Bond et al., 2005*; *Gul et al., 2006*; *Darvish et al., 2010*). CPAP and STIL are strictly required for centriole assembly: STIL at a very early stage (*Stevens et al., 2010b*; *Tang et al., 2011*; *Kitagawa et al., 2011a*; *Arquint et al., 2012*; *Vulprecht et al., 2012*) and CPAP slightly later (*Kirkham et al., 2003*; *Leidel and Gonczy, 2003*; *Basto et al., 2006*; *Kleylein-Sohn et al., 2007*; *Dobbelaere et al., 2008*; *Vulprecht et al., 2012*), possibly by

controlling the organisation (*Pelletier et al., 2006*; *Dammermann et al., 2008*) and length of the centriolar microtubules (*Blachon et al., 2009*; *Kohlmaier et al., 2009*; *Schmidt et al., 2009*; *Tang et al., 2009*; *Kim et al., 2012*). A direct interaction between STIL and CPAP has been observed in yeast-two-hybrid and pull-down experiments (*Tang et al., 2011*; *Vulprecht et al., 2012*). Intriguingly, a MCPH mutation (E1235V) in the conserved C-terminal domain of CPAP (the so-called TCP-domain or G-Box) appeared to weaken this yeast-two-hybrid interaction (*Tang et al., 2011*). Tissue culture experiments suggested that this MCPH mutation might cause a partial loss-of-function of CPAP (*Kitagawa et al., 2011a*). However, the same study also found that the E1235V mutation results in an enhanced functionality of CPAP when overexpressed in vivo (*Kitagawa et al., 2011a*). To understand how CPAP and STIL interact and how the MCPH mutation affects CPAP functionality in vitro and in vivo, we undertook a detailed biochemical, structural and functional study of the putative CPAP–STIL complex.

## Results

### The CPAP TCP domain binds to a conserved proline-rich motif in STIL

Yeast-two-hybrid experiments suggested that a region of human CPAP comprising its conserved C-terminal TCP domain (or G-box) can interact with a ~400 amino acid (aa) region (residues 231–619) of human STIL (*Tang et al., 2011*). To try to identify the region of STIL most likely to be involved in an interaction with CPAP, we carried out a sequence alignment with multiple metazoan STIL proteins (*Figure 1A*, *Figure 1—figure supplement 1*). This analysis revealed a short (~40 aa) highly conserved proline-rich region (CR2) (*Figure 1A*) within this interval. To test whether this region of STIL could bind to the CPAP TCP domain, we recombinantly produced the TCP domain of *Danio rerio* CPAP and used isothermal titration calorimetry (ITC) to test its ability to bind to a fragment of *D. rerio* STIL that spanned CR2 (residues 404–448) (*Figure 1D*, *Table 1*). The two proteins formed a 1:1 complex with a $K_D$ of ~2 μM. Next, we further split the peptide to test the binding contribution from its N-terminal (residues 411–428) and C-terminal region (residues 429–448). The N-terminal region exhibited an only slightly weaker binding ($K_D$ ~4 μM) to the TCP domain, whereas the C-terminal region showed a very weak binding ($K_D$ > 500 μM) (*Figure 1D*; *Table 1*). We conclude that the CPAP TCP domain binds to a short conserved motif in STIL (CR2) with a potentially biologically significant affinity, and that the majority of the binding affinity comes from interactions with residues within the first proline-rich region in CR2.

### The CPAP TCP domain adopts a unique extended open β-sheet conformation that packs against a series of conserved prolines in STIL

To understand how CPAP and STIL interact at the molecular level, we obtained the crystal structures of the TCP-domain of *D. rerio* CPAP[937–1124], both on its own and in a complex with *D. rerio* STIL[408–428] (*Figure 1B,C,E*; *Table 2*, *Table 3*). In both structures, the TCP domain adopts a nearly identical conformation, suggesting that no significant conformational change occurs in CPAP upon binding to STIL (RMSD = 1.5 Å ± 0.2 Å over 148 ± 4 Cα pairs). The TCP domain folds into a single layer β-sheet comprising ~20 consecutive antiparallel strands connected by type I β-turns and is stabilised by an extensive hydrogen-bonding network. The resulting sheet shows a twist of approximately 13° (i.e., the angle between the consecutive, hydrogen-bonded strands), slightly lower than the average value of 20° observed for typical β-sheets (*Chothia, 1973*; *Murzin, 1992*). Individual β-hairpins correspond to previously noted (*Islam et al., 1993*; *Hung et al., 2000*) repeats in the TCP domain sequence; the turns of these hairpins are often constituted by a PDG motif explaining the high frequency of proline and glycine residues in this domain (*Figure 1—figure supplement 2*). Crystal packing interactions involve only small protein interfaces, suggesting that the protein is biologically active as a monomer. Indeed both small-angle X-ray scattering (SAXS) and size-exclusion chromatography—multi angle light scattering (SEC-MALS) experiments demonstrate that the TCP domain is predominantly monomeric in solution (*Figure 1—figure supplement 3*).

The structure of the TCP domain represents an unusual, novel architecture. It is reminiscent of the β-sheet conformation proposed to exist within amyloid fibrils and resembles engineered water-soluble peptide self-assembly mimics (PSAMs) used to study β-rich self-assemblies (*Makabe et al., 2006*). In contrast to these PSAM structures whose conformation is maintained by two globular domains capping both ends of the β-sheet, the TCP domain stably exists on its own. (*Figure 1—figure supplement 4*). The TCP domain structure lacks a defined hydrophobic core typical for globular domains, and both sides of its β-sheet are exposed to the solvent and well hydrated.

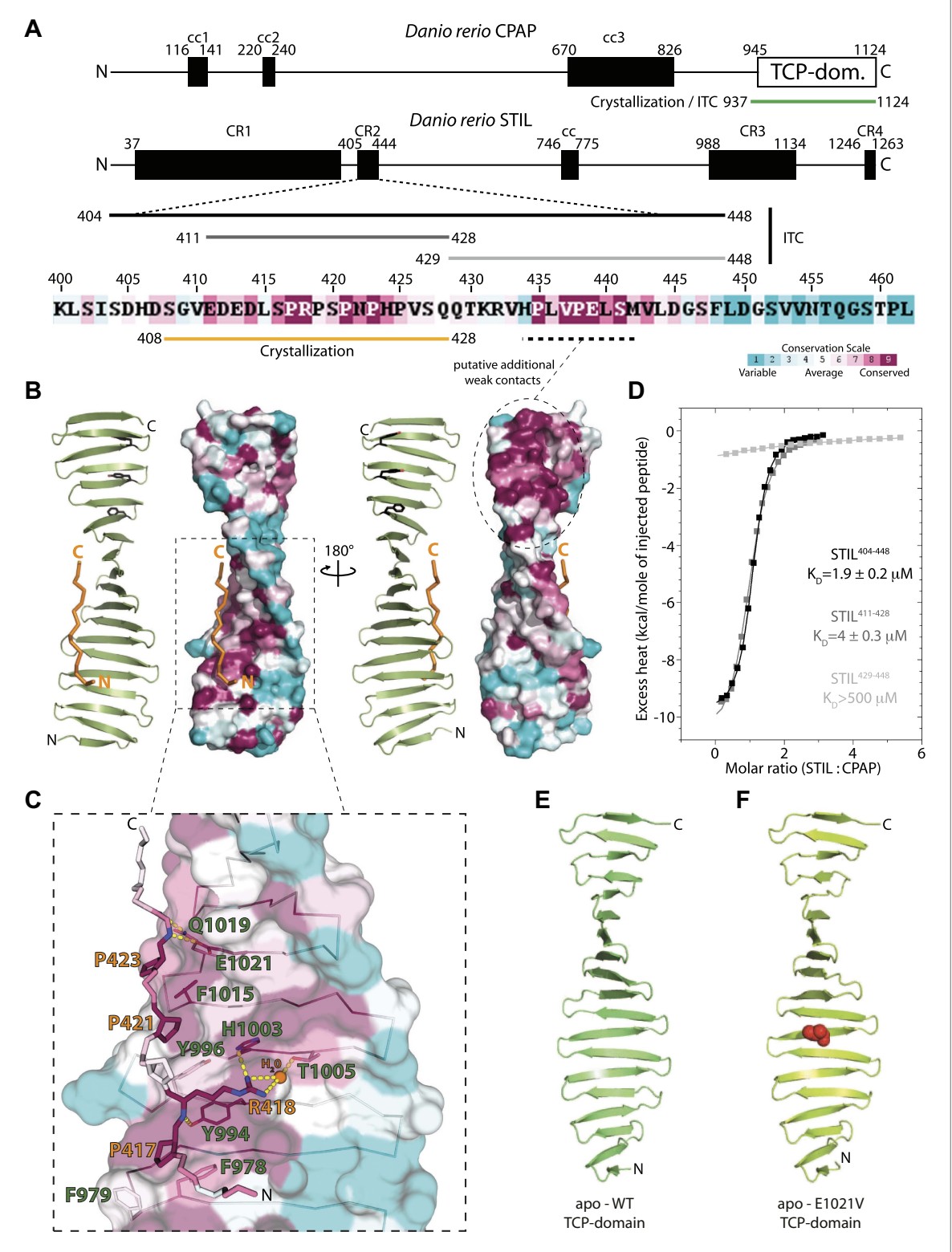

**Figure 1**. Biochemical and structural characterisation of the CPAP TCP domain and its interaction with STIL. (**A**) Schematic representation of *D. rerio* CPAP and STIL. CPAP is a 1124 amino acid (aa) protein with three predicted coiled coil (cc) domains and a C-terminal TCP domain. STIL is a 1263 aa protein with one predicted cc domain and several conserved regions (CR). The proline-rich CR2 domain is enlarged and coloured according to Consurf conservation scores (**Glaser et al., 2003**) from cyan (variable) to burgundy (conserved). The constructs used in this study are indicated by bars.
*Figure 1. Continued on next page*

*Figure 1. Continued*

(**B**) Two views of the TCP domain structure (green) in complex with the STIL peptide (orange), rotated by 180°. Images on the left of each view show a ribbon representation and images on the right show the TCP domain as a molecular surface coloured according to Consurf conservation scores. Note the presence of a conserved patch (dashed circle) along the edge of the TCP domain where the STIL peptide is bound. This patch contains aromatic residues (black sticks) that would be well placed to interact with conserved prolines in the C-terminal part of the STIL CR2 region that we had to omit for crystallisation. ITC experiments (*Figure 1D*) suggest that these putative additional contacts would only contribute weakly to overall binding. (**C**) Detailed view of the *D. rerio* CPAP–STIL interaction interface coloured according to Consurf conservation scores. Interface residues are shown in sticks, and the TCP domain is shown as a semi-transparent molecular surface. Contact residues are labelled in green (CPAP) and orange (STIL). Dotted yellow lines indicate hydrogen-bonds. The dark orange sphere represents a bound water molecule. (**D**) ITC analysis using the STIL constructs shown in *Figure 1A*. The excess heat measured on titrating STIL into CPAP at 25°C was fitted to a single set of binding sites model. Fitted $K_D$ values are indicated together with their standard deviations. (**E** and **F**) Ribbon models of the apo-structures of the *D. rerio* CPAP TCP domain: (**E**) WT apo-structure; (**F**) E1021V (MCPH mutation) apo-structure (V1021 represented as red spheres).

The following figure supplements are available for figure 1:

**Figure supplement 1**. Multiple sequence alignment of the conserved proline-rich region of STIL (CR2).

**Figure supplement 2**. The TCP domain sequence repeats.

**Figure supplement 3**. The TCP domain of CPAP is predominantly monomeric in solution.

**Figure supplement 4**. The TCP domain resembles engineered peptide-assembly mimics used to study β-rich self-assemblies.

**Figure supplement 5**. Multiple sequence alignment of the TCP domain of CPAP.

The structure of the CPAP–STIL complex revealed that the STIL peptide binds in a polyproline II helical conformation along one edge of the TCP domain β-sheet. The STIL peptide binds to CPAP by four main mechanisms (*Figure 1C*). First, three STIL prolines (P417, P421, and P423) pack against aromatic CPAP residues (F978, Y996, and F1015) in a way that resembles target motif recognition by other described proline-rich motif (PRM) binding domains (*Kay et al., 2000*). Second, R418 (STIL) makes a cation-π interaction with the phenyl ring of Y994 (CPAP). Third, STIL R418 is further involved in a water-mediated hydrogen bonding network that includes CPAP residues H1003 and T1005. Finally, sidechain–mainchain interactions are formed between CPAP residues Y994, Q1019, and E1021 and the bound STIL peptide. The CPAP and STIL residues involved in this interaction are highly conserved across metazoans (*Figure 1C*).

Sequence conservation of the TCP domain is not confined to this section of our structure but extends further along the same edge of the sheet (*Figure 1B*). This additional conserved region contains aromatic residues that are arranged similar to those that pack against the proline residues of the bound STIL peptide in our crystal structure (*Figure 1B,C*). Intriguingly, the C-terminal part of STIL's CR2 region (omitted to obtain diffraction grade crystals) contains two highly conserved proline

**Table 1.** Characterisation of the CPAP:STIL interaction in vitro

| *Danio rerio* STIL peptide in syringe | *Danio rerio* CPAP[937–1124] TCP domain in cell | Number of binding sites (N) | SD N | $K_D$ (µM) | SD $K_D$ (µM) | ΔH (kcal/mol) | SD (kcal/mol) | n (number of measurements) | Factor change in $K_D$ |
|---|---|---|---|---|---|---|---|---|---|
| STIL[404–448] | WT | 1.07 | 0.04 | 1.9 | 0.2 | −10.1 | 0.3 | 5 | 1 |
| STIL[411–428] | WT | 0.98 | 0.03 | 4 | 0.3 | −11.3 | 0.5 | 3 | 2 |
| STIL[429–448] | WT | 0.97 | 0.08 | 540 | 130 | −6.3 | 0.6 | 2 | ~280 |

Binding parameters between *D. rerio* CPAP and various *D. rerio* STIL constructs obtained from ITC experiments. Fitting was performed with N as a variable. Constraining N to a fixed value of 1 during fitting produced $K_D$ values that were within the experimental error of those tabulated here.

residues (P435 and P438 in *D. rerio*) that would be well positioned to bind to these aromatic residues in an analogous way (*Figure 1A,B*). Thus, we speculate that the entire CR2 region of STIL spanning from residue 417 to residue 438 (*D. rerio*) may be bound all along the edge of the TCP domain. Although our ITC experiments suggest that these putative additional contacts are insufficient to establish strong binding between STIL and CPAP (*Figure 1D*) they may contribute cooperatively to the CPAP–STIL interaction once the N-terminal proline-rich region in CR2 established binding. We conclude that the TCP domain of CPAP adopts a unique extended open β-sheet conformation that recognises a series of conserved prolines in the CR2 region of STIL.

## The CPAP E1021V MCPH mutation reduces the binding affinity of the CPAP–STIL interaction

The involvement of CPAP E1021 in the interaction with STIL in zebrafish is potentially significant, as the equivalent residue in human CPAP (E1235) is mutated to valine in some MCPH patients. To test whether this mutation disrupts the organisation of the TCP domain, we obtained the crystal structure of *D. rerio* CPAP$^{937–1124}$ carrying the E1021V mutation (*Figure 1F*; *Table 2*). The structure of the wild-type and the mutant TCP domain were virtually identical (RMSD = 0.1 Å over 142 Cα pairs) demonstrating that the TCP domain structure was not compromised. To test whether this mutation perturbed the interaction with STIL, we purified WT and various other mutant forms of *D. rerio* CPAP$^{937–1124}$ in which we valine substituted residues that our crystal structure suggested to be important for binding (*Figure 2B*). Circular dichroism (CD) spectra indicated that the mutant forms of the TCP domain were correctly folded with a predominantly β-type profile (*Figure 2—figure supplement 1*). ITC experiments with WT *D. rerio* STIL$^{404–448}$ showed that the mutation of residues F978, Y994, and F1015 decreased the binding strength by ~20 to 40-fold (*Figure 2A*, left; *Table 4*), while mutation of E1021 decreased the binding strength by approximately eightfold. In contrast, mutation of T986, which is not predicted to be in the interaction interface, did not detectably perturb binding.

We also purified mutant forms of the *D. rerio* STIL$^{404–448}$ peptide and tested their ability to interact with the WT *D. rerio* TCP domain in ITC experiments (*Figure 2A*, right, *Figure 2—figure supplement 1C*; *Table 4*). Alanine substitution of P417, R418 or P421 decreased the binding strength by ~10 to 20-fold and alanine substitution of P423 by approximately twofold to threefold. In contrast, the mutation of residue N422, which is not predicted to be in the interaction interface, did not compromise binding. Taken together these results lend strong support to our structural model and indicate that the E1021V MCPH mutation leads to roughly an order of magnitude decrease in affinity of the CPAP–STIL interaction.

## The CPAP–STIL interaction is highly conserved

The sequence conservation of the CPAP TCP domain (*Figure 1—figure supplement 5*) and the CR2 region of STIL (*Figure 1—figure supplement 1*) suggests that this interaction may be conserved. To confirm this, we solved the crystal structure of the TCP domain from *Drosophila melanogaster* DSas-4 (dCPAP) (residues 700–901) in complex with the region of Ana2 (dSTIL) equivalent to CR2 (residues 1–47) (*Table 2*, *Table 5*, *Table 6*; *Figure 2C*). The dSTIL–dCPAP interaction interface in this structure was highly similar to the *D. rerio* complex (inter-species alignments of the structures yielded an average pairwise RMSD of 1.2 ± 0.2 Å across an average of 118 ± 4 Cα pairs). Indeed, all copies of the complex obtained in the structures from both species superimposed well and exhibited the same four major groups of binding interactions as described for the *D. rerio* structure. This conservation includes the contact made by the E792 residue in dCPAP (the equivalent of the E1235 residue in human CPAP that is mutated in MCPH). Together, these data allow us to determine a consensus CPAP binding motif in metazoan STIL proteins (PRxxPxP, *Figure 1—figure supplement 1*) and suggest that the described CPAP–STIL interaction constitutes a highly conserved step in centriole biogenesis.

## The CPAP–STIL interaction is essential for centriole assembly in vivo

Since the binding mechanism of CPAP and STIL is conserved between zebrafish and *Drosophila*, we turned to *D. melanogaster* as a model system to address the functional relevance of this interaction in vivo. In flies, the lack of dCPAP or dSTIL leads to centriole loss and a consequent severe uncoordinated (unc) phenotype due to the lack of basal bodies and so cilia in Type I sensory neurons. These flies lack all mechano- and chemo-sensation and, although viable, they usually die shortly after eclosion, as they

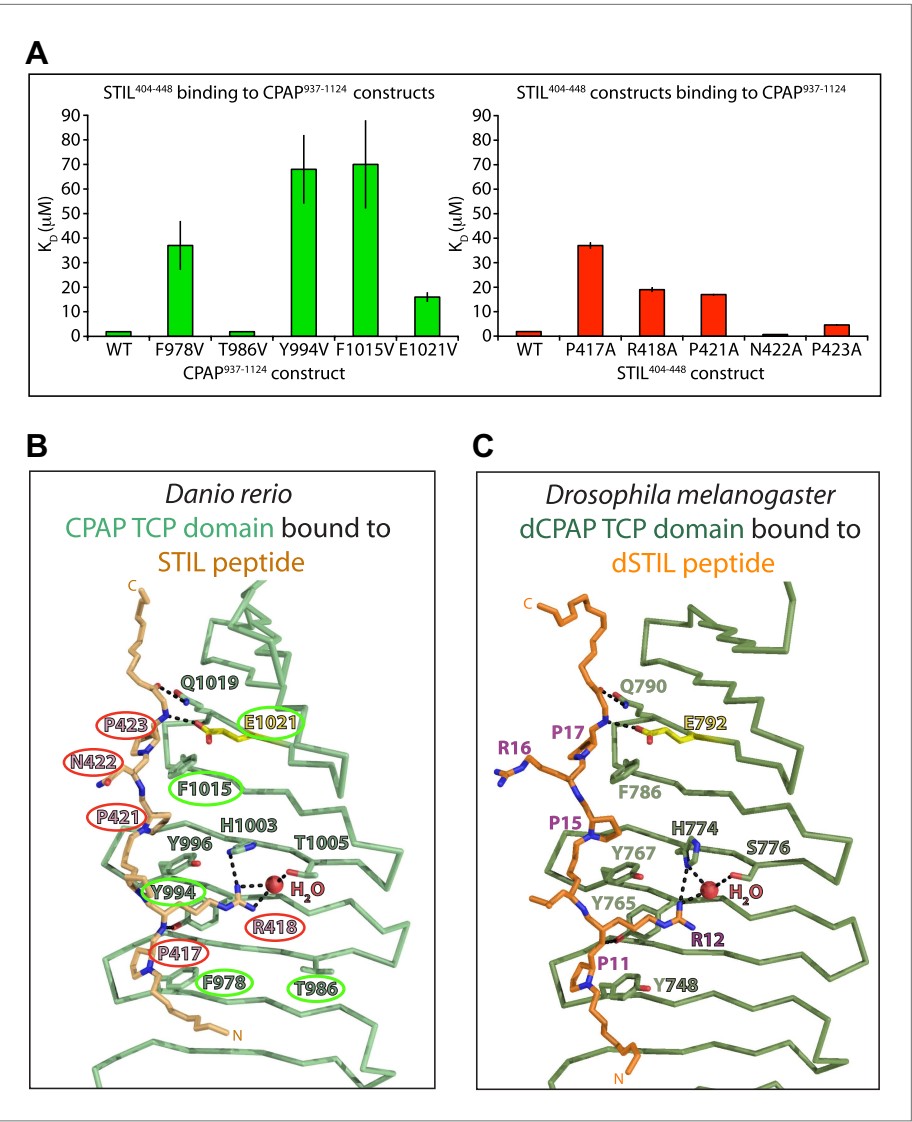

**Figure 2.** Mutational analysis of the CPAP:STIL interaction in vitro and conservation of the interaction across species. (**A**) Graphs showing the binding constants ($K_D$) determined by ITC for the interaction between WT and mutant constructs of CPAP[937–1124] and STIL[404–448]. Left panel, WT and various mutant forms of CPAP[937–1124] binding to WT STIL[404–448] (T986 is a non-interacting residue included as a negative control). Error bars, standard deviation. Right panel, WT and various mutant STIL[404–448] constructs binding to WT CPAP[937–1124] (N422 is a non-interacting residue included as a negative control). Error bars, standard deviation. The wild-type measurements are the same as shown in **Figure 1D** and are shown again for comparison to the mutants. (**B** and **C**) Close-up view of the CPAP (green):STIL (orange) interaction interface from *D. rerio* (**B**) and *Drosophila* (**C**). Interface residues are shown as sticks, in yellow is the Glutamate residue in *Drosophila* and *D. rerio* CPAP that is equivalent to E1235 in human CPAP (mutated in MCPH). Residues of the *D. rerio* protein mutated for ITC experiments are ringed in green (CPAP) or red (STIL). Dotted black lines indicate hydrogen-bonds. The conserved bound water molecule is shown as a red sphere.

The following figure supplements are available for figure 2:

**Figure supplement 1**. Characterisation of the *D. rerio* TCP domain mutants and STIL peptide mutants used for thermodynamic analysis.

cannot feed or move in a coordinated fashion (**Kernan et al., 1994**; **Basto et al., 2006**; **Wang et al., 2011**). We examined the ability of various GFP-tagged versions of dCPAP and dSTIL to rescue the centriole loss observed in these mutants and assayed their ability to localise to centrosomes in the presence of endogenous dCPAP or dSTIL (**Figure 3**).

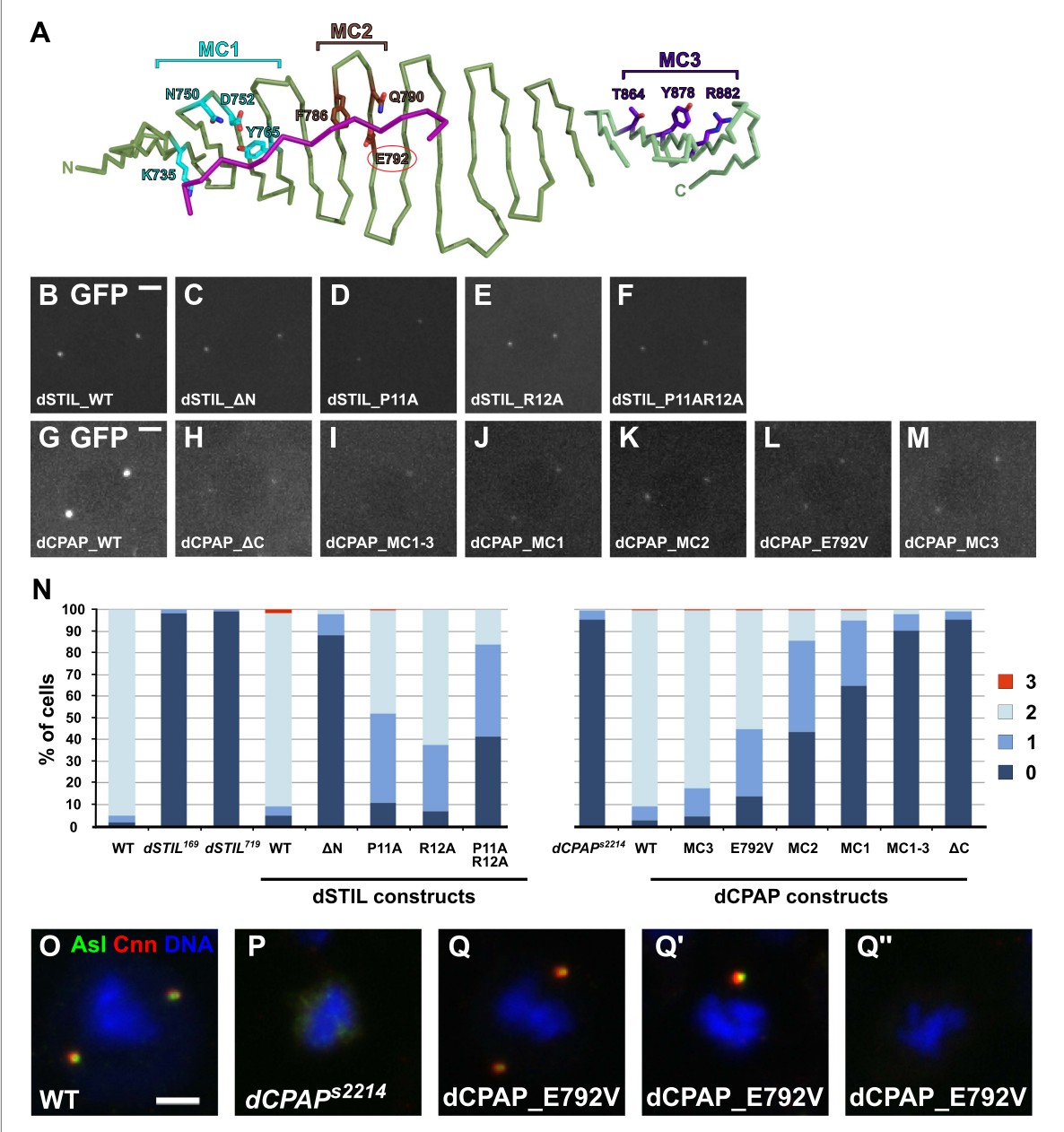

**Figure 3**. The interaction between dCPAP and dSTIL is essential for centriole duplication in *Drosophila*. (**A**) Schematic view of the complex between dCPAP (green) and dSTIL (magenta) with the residues mutated in MC1 (cyan), MC2 (brown) and MC3 (dark purple) indicated as coloured sticks. The MCPH residue E792 is circled in red. Note that MC1 and MC2 are mapped onto the *Drosophila* structure (dark-green backbone), while MC3 had to be mapped onto the backbone of the *D. rerio* structure (light green backbone). Although highly conserved between *Drosophila* and *D. rerio* (*Figure 1—figure supplement 5*) this region was not visible in the electron density map of the *Drosophila* structure probably due to its partial unfolding to enable packing interactions within the crystal. (**B–M)** Panels show representative still images taken from movies of *Drosophila* embryos expressing the indicated dCPAP-GFP or dSTIL-GFP constructs. Note that all analyses were performed in the presence of endogenous WT dCPAP or dSTIL, and that all images were acquired with the same microscope settings at the same stage of the cell cycle. (**B–F**) dSTIL-GFP constructs localise to centrosomes at similar levels. (**G–M**) All mutant dCPAP-GFP constructs localise to centrosomes, but at strongly reduced levels compared to wild-type dCPAP-GFP. (**N**) Graphs show the percentage of cells with 0, 1, 2, and 3 centrosomes in the genotypes analysed (as indicated). All dSTIL-GFP and dCPAP-GFP constructs were analysed in their respective mutant backgrounds. Note that this experiment was performed blind. (**O–Q''**) Panels show third instar larval brain cells of various genotypes in metaphase. Cells were stained for the centriolar protein Asterless (Asl—green) and the PCM component Centrosomin (Cnn—red) and DNA (blue). Wild-type metaphase cells have two centrosomes (**O**), whereas centrosomes are mostly absent in third instar larval brain cells from *dCPAP* mutants (**P**). As an example, representative images of *dCPAP* mutant cells expressing the dCPAP_E792V-GFP construct are shown that were scored with 2 (**Q**), 1 (**Q'**) or no (**Q''**) centrosomes. Scale bars = 3 µm.

*Figure 3. Continued on next page*

*Figure 3. Continued*

The following figure supplements are available for figure 3:

**Figure supplement 1**. Protein expression levels of GFP-tagged dCPAP and dSTIL constructs in *dCPAP* or *dSTIL* mutant *Drosophila* brain cells and quantification of their centriole/centrosome numbers.

We first expressed a version of dSTIL that lacks the first 45 aa (including the PRxxPxP motif required for the interaction with dCPAP). GFP-tagged wild-type dSTIL (dSTIL_WT-GFP) served as a control. Both proteins were expressed at similar levels and localised strongly to centrosomes in the presence of endogenous dSTIL (*Figure 3B,C*; *Figure 3—figure supplement 1*). Only the wild-type version, however, was able to rescue the unc phenotype and the centriole loss phenotype of the *dSTIL* mutant (*Figure 3N*; *Table 7*). To further characterise the dCPAP binding domain in vivo we mutated the first proline and arginine of the PRxxPxP motif of dSTIL to alanine, both separately and in combination (P11A, R12A, and P11A:R12A, *Figure 2C*). All three constructs strongly localised to centrosomes in the presence of endogenous dSTIL (*Figure 3D–F*, *Figure 3—figure supplement 1*). Both single mutants rescued the unc phenotype of the *dSTIL* mutation while the double mutant failed to do so (data not shown). The single mutants P11A and R12A were also able to partially rescue the centriole loss phenotype, whereas the double mutant P11A:R12A showed only a poor rescue (*Figure 3N*; *Table 7*). These data strongly suggest that the interaction with dCPAP is essential for dSTIL function in centriole assembly.

We next deleted the entire TCP-domain of dCPAP (dCPAP_ΔC), or expressed GFP fusion proteins carrying mutation clusters (MCs) altering 3–4 residues in different regions of the TCP domain (*Figure 3A*). Mutation clusters were designed that targeted central (dCPAP_MC1) or peripheral (dCPAP_MC2) residues in the dSTIL binding domain, as well as residues that are predicted to not significantly be involved in complex formation (dCPAP_MC3), according to the crystal structure and the ITC data (*Figure 3A*, *Figure 2A*, *Figure 1D*; *Table 1*). We also analysed dCPAP_E792V-GFP lines, which carried the MCPH equivalent mutation E792V (E1235V in humans and E1021V in zebrafish CPAP). All transgenic dCPAP-GFP proteins were expressed at approximately equivalent levels in vivo, but were moderately overexpressed compared to endogenous dCPAP (*Figure 3—figure supplement 1*). Wild-type dCPAP-GFP localised strongly to centrosomes and rescued both the unc phenotype and the centriole loss phenotype (*Figure 3G,N*; *Table 7*). Strikingly, the rescuing ability of the mutant constructs strongly correlated with the predicted strength of dSTIL binding. dCPAP_ΔC-GFP failed to rescue, dCPAP-MC1 and dCPAP-MC2 rescued poorly, the MCPH mutation E792V showed an intermediate phenotype, while dCPAP-MC3 exhibited a robust rescue (*Figure 3N*; *Table 7*). Interestingly, when compared to wild-type dCPAP-GFP, all mutant constructs (including dCPAP_MC3) localised only weakly to centrosomes (*Figure 3H–M*). Together, these data suggest that the interaction between dCPAP and dSTIL is a key step in centriole assembly and is essential for centriole duplication. Furthermore, they indicate that low total levels of dCPAP at centrosomes might be sufficient for centriole duplication, as long as some interaction with dSTIL is maintained.

## The TCP domain of *C. elegans* SAS-4 is required for its interaction with SAS-5 and for centriole assembly

It has been proposed that SAS-5 is the *C. elegans* homolog of the STIL proteins in flies and vertebrates, but there is little sequence homology between these proteins (*Stevens et al., 2010a*). We failed to identify an unambiguous PRxxPxP motif in worm SAS-5, so we tested whether the TCP domain of SAS-4 (the *C. elegans* CPAP homologue) is functionally important. We used the Mos single-copy insertion system (MosSCI; *Frøkjær-Jensen et al., 2008*) to generate transgenic lines with single-copy transgenes under the control of *sas-6* regulatory sequences integrated at a specific site on chromosome II (*Figure 4A*). Transgenes were generated expressing GFP fusions with either WT SAS-4 (SAS-4^WT::GFP) or a form in which the C-terminal TCP domain (aa 557–808) had been deleted (SAS-4^ΔTCP::GFP); both transgenes contained a 497 bp resequenced region in their N-terminal coding region (preserving codon usage) that rendered them resistant to RNAi-mediated depletion (*Figure 4A*).

SAS-4 depletion by RNAi prevents centriole assembly, resulting in a signature phenotype characterised by a normal first mitotic division followed by monopolar spindles during the second division (*Figure 4B,C*; *O'Connell et al., 2001*). This phenotype arises because the sperm that fertilise the

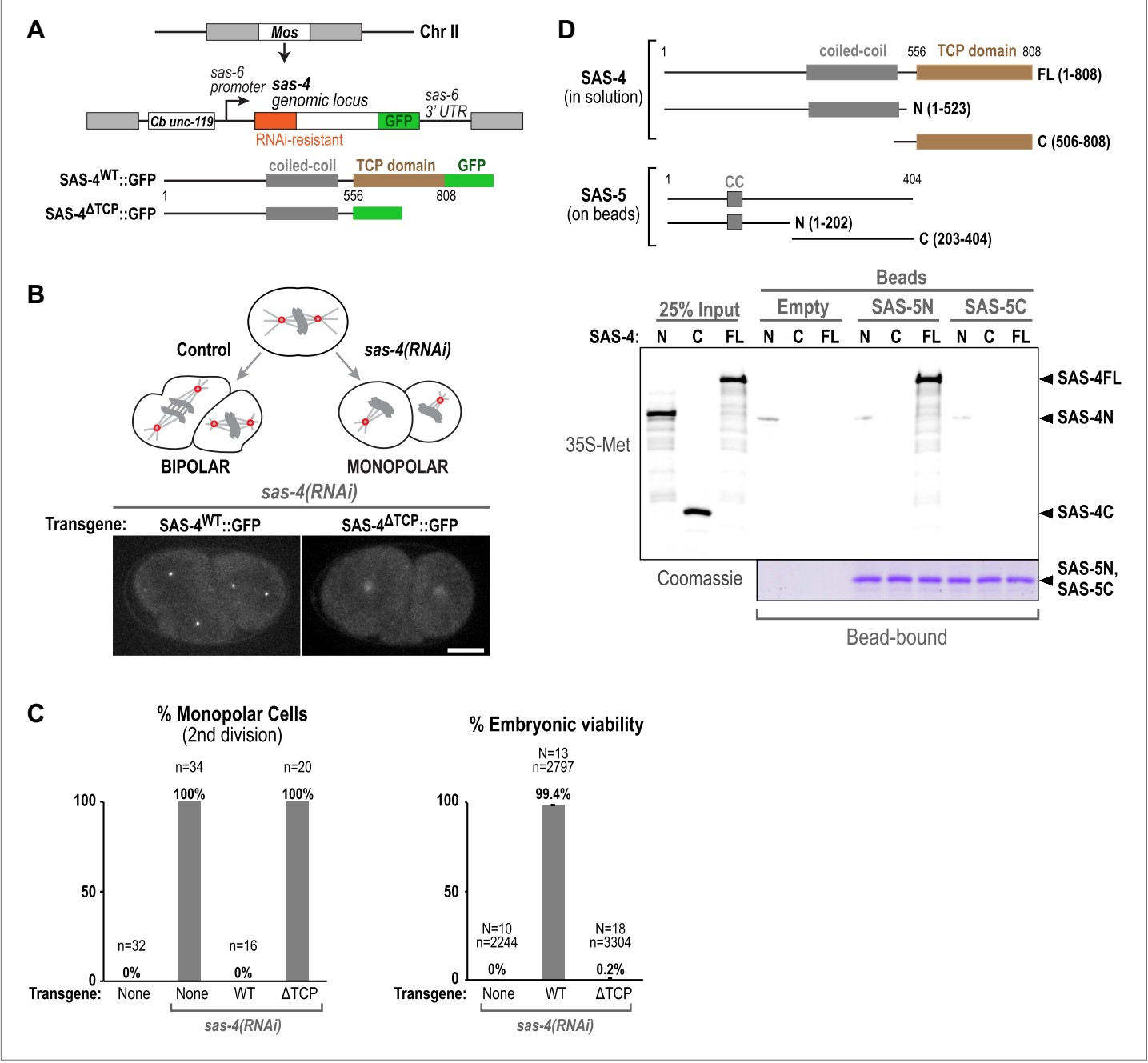

**Figure 4**. The TCP domain of *C. elegans* SAS-4 is required for its interaction with SAS-5, its localisation to centrioles, and for centriole assembly. (**A**) Schematic illustration of the MosSCI system used for generating single-copy *sas-4* transgene insertions. (**B**) A schematic illustration of the monopolar spindle assay for centriole duplication in *C. elegans* embryos. Panels show maximum intensity projections of representative fluorescence confocal z-series taken of *sas-4(RNAi)* embryos expressing either WT or ΔTCP SAS-4::GFP. Transgenic SAS-4^WT^::GFP localises to sharp foci representing the centrioles, whereas SAS-4^ΔTCP^::GFP localises diffusely to the pericentriolar material. Bar, 10 μM. (**C**) Graphs show the quantification of second division monopolar spindles (left) and embryonic viability (right) after *sas-4(RNAi)* and rescue with either a WT or ΔTCP *sas-4::gfp* transgene. (**D**) Panels show autoradiographs (top panel) and a Coomassie stained gel from a Ni-NTA pull-down experiment with $^{35}$S-labelled in vitro translated SAS-4 fragments (prey) and SAS-5-6xHis fragments (baits).

SAS-4-depleted oocytes carry two normal centrioles, since sperm are produced prior to introduction of the dsRNA. The two sperm-derived centrioles organise two centrosomes, and the first mitotic division appears normal. If no new centrioles form, each daughter cell inherits a single sperm-derived centriole leading to monopolar spindles in the second division and 100% embryonic lethality

(*Figure 4C*). Both the monopolar spindle phenotype and embryonic viability were fully rescued by the WT *sas-4::gfp* transgene (aa 1–808), but not by the ΔTCP transgene (aa 1–556) (*Figure 4B,C*). While SAS-4[WT]::GFP targeted to centrioles in the absence of the endogenous protein, SAS-4[ΔTCP]::GFP did not, and instead exhibited a diffuse accumulation in the pericentriolar material (*Figure 4B*). Thus, the SAS-4 TCP domain is required for SAS-4 to accumulate at centrioles and become incorporated into the microtubule-containing outer centriole wall.

To determine if the failure of SAS-4[ΔTCP]::GFP to become incorporated in the centriole outer wall could be due to an inability to interact with SAS-5, we performed a pull-down assay to determine whether [35]S-labelled in vitro translated SAS-4 fragments could interact with the N-terminal or C-terminal regions of SAS-5 bound to beads (*Figure 4D*). In vitro translated full-length SAS-4 interacted specifically with the N-terminal domain (aa 1–202) of SAS-5. Interestingly, we could not further narrow down the region of SAS-4 required for this interaction. Neither the SAS-4 N-terminal nor C-terminal region (which includes the TCP domain) alone could be pulled down by SAS-5. This result suggests that although the TCP domain is required for SAS-4 to interact with SAS-5, it is not sufficient. Together, these data suggest that a TCP domain-dependent interaction between SAS-4/CPAP and SAS-5/STIL is conserved and essential for centriole duplication in *C. elegans*, but that the precise interaction interface may have diverged.

## Discussion

Only a small set of conserved centriolar proteins is essential for centriole assembly (*Brito et al., 2012*; *Gonczy, 2012*) and some of these proteins, like CPAP and STIL, have been linked to microcephaly in humans (*Leal et al., 2003*; *Bond et al., 2005*; *Gul et al., 2006*; *Thornton and Woods, 2009*; *Darvish et al., 2010*). However, there is currently little structural understanding on how these proteins interact with one another, how mutations in them cause microcephaly in humans and how these interactions are regulated.

Here we have solved the crystal structures of the CPAP–STIL complex from zebrafish and *Drosophila*. We showed that the CPAP TCP domain folds into an elongated open-sided β-meander that consists of ~20 consecutive antiparallel β-strands connected by type I β-turns. β-meanders are frequently found in β-barrels, β-propellers and some α+β proteins. However, what, to our knowledge, makes the TCP domain structure unique amongst naturally occurring proteins is that it solely consists of a freestanding meander β-sheet that entirely lacks a defined hydrophobic core and is not flanked by other globular domains that pack against it. We show that the TCP domain is predominantly monomeric in solution and self-interacts in its crystallised form only through small interfaces that are not conserved. Thus, despite some reminiscence to β-sheets observed in amyloid fibrils it is unlikely that the TCP domain self-associates in a similar manner.

Instead, we demonstrate that the TCP domain of CPAP constitutes a novel proline-rich-motif (PRM) recognition-domain that specifically binds to a short target motif in STIL. Although the overall sequence identity of the CPAP and STIL proteins between *Drosophila* and zebrafish is relatively low (~22% and ~13%, respectively), our structural analysis revealed that the interaction interface is conserved, confirming the previous proposal that fly Ana2 is the functional homologue of vertebrate STIL (*Stevens et al., 2010a*). Our characterisation of the binding interface also allowed us to define a consensus-binding site (PRxxPxP) for the CPAP TCP domain in STIL that is conserved across metazoa. Our mutational analysis of the interface demonstrates a remarkable correlation between the ability of mutant proteins to bind to one another in vitro and their ability to support centriole assembly in vivo, providing compelling support for our structural model of the metazoan CPAP–STIL complex. These data strongly suggest that the interaction between CPAP and STIL is a conserved, essential step in centriole biogenesis. A schematic model that places this interaction in the context of a possible centriole assembly pathway is shown in *Figure 5*.

The high degree of sequence divergence between vertebrate STIL, *Drosophila* Ana2 and *C. elegans* SAS-5 suggests that STIL homologs are under particularly strong lineage-specific selection. Despite the many sequence changes between *Drosophila* Ana2 and vertebrate STIL, our work suggests that the interaction interface between Ana2/STIL and dSAS-4/CPAP TCP domain has been retained, highlighting its importance. Even in *C. elegans*, which is the most divergent of the functionally characterised STIL homologs, our work indicates that the SAS-4 TCP domain is essential for centriole assembly, and that a TCP-domain dependent interaction between SAS-4 and SAS-5 has been conserved. Nevertheless, as the SAS-4 TCP domain is not sufficient for interaction with SAS-5 and we were unable to identify a PRxxPxP interaction motif in worm SAS-5, more work will be needed to understand

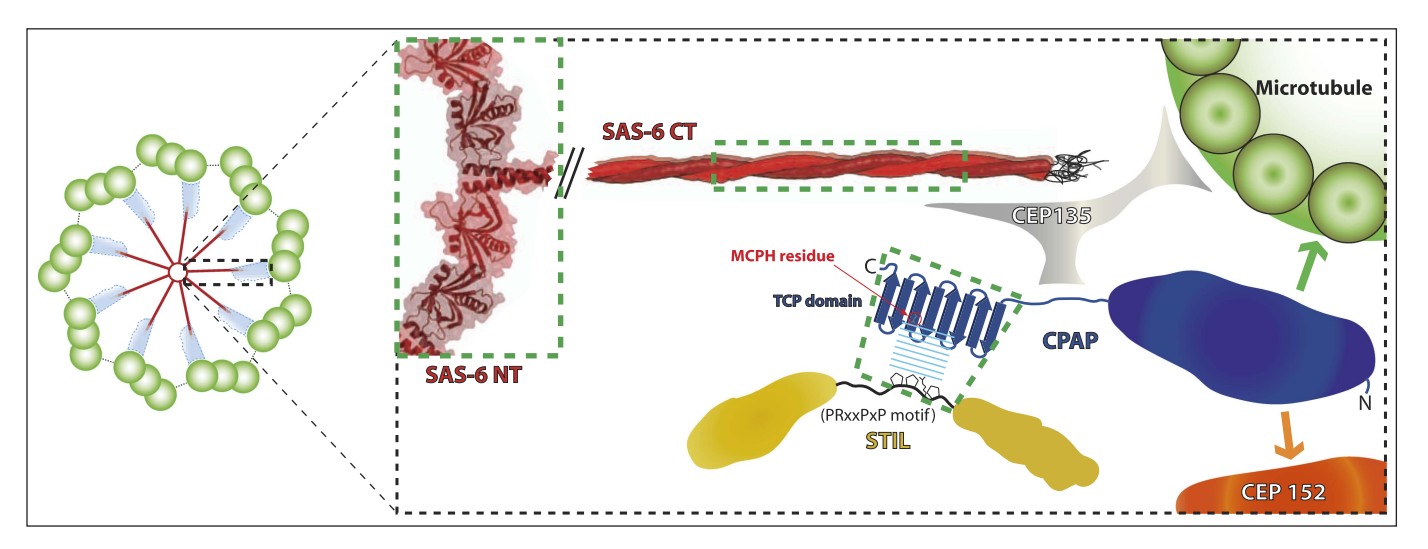

**Figure 5**. A schematic representation of protein interactions within the inner region of the centriole. In this illustration, interactions whose crystal structure have been determined are highlighted by green boxes—all other interactions are inferred from biochemical and genetic studies and so are depicted in cartoon form. The cartwheel central hub comprises SAS-6 (red) (*Nakazawa et al., 2007*; *Kitagawa et al., 2011b*; *van Breugel et al., 2011*). The spokes extending outward from the hub consist of a homodimeric SAS-6 coiled-coil, which extends (*van Breugel et al., 2011*) into a region known as the 'pinhead' (cyan in low magnification view, left), where CEP135 (grey) may act as a linker between SAS-6, CPAP and microtubules (*Hiraki et al., 2007*; *Roque et al., 2012*; *Lin et al., 2013*). CPAP (dark blue) localises more towards the periphery of the centriole (*Mennella et al., 2012*; *Sonnen et al., 2012*; *Lukinavičius et al., 2013*), where its N-terminal part may interact directly with both Asterless/CEP152 (*Cizmecioglu et al., 2010*; *Dzhindzhev et al., 2010*) (orange arrow) and microtubules (*Hsu et al., 2008*) (green arrow). In contrast STIL (yellow) localises more towards the interior of centrioles (*Arquint et al., 2012*), and appears to function upstream of CPAP in centriole biogenesis (*Tang et al., 2011*; *Vulprecht et al., 2012*). Thus, we propose that the C-terminal TCP domain of CPAP interacts with the conserved region 2 (CR2) of STIL towards the interior of the centriole and that this interaction is crucial for CPAP/STIL function at centrioles. The orientation of STIL in centrioles is unknown.

the SAS-4—SAS-5 interaction in *C. elegans* and its relationship to the CPAP–STIL interaction in other metazoans.

A surprising aspect of our findings is that the E792V (MCPH) mutant and all three of the mutation clusters (MCs) that we analysed in dCPAP localise poorly to centrosomes. For the E792, MC1, and MC2 mutations this could be expected, as these are all predicted to perturb the interaction between dCPAP and dSTIL (as is the case with similar mutations in zebrafish CPAP in our in vitro binding assays), and this would be predicted to perturb the recruitment of dCPAP to centrioles. The MC3 cluster, however, is not predicted to lie in a strong interaction interface and, unlike the MC1 and MC2 mutation clusters, it can rescue the centriole duplication defect in dCPAP mutant cells nearly as efficiently as the WT protein. Possibly, an interaction with another protein that plays some part in recruiting dCPAP to centrioles might be perturbed by these mutations. Alternatively, similar to the situation with *C. elegans* SAS-4 (*Dammermann et al., 2008*), dCPAP may localise to both centrioles and the PCM. It might therefore be PCM and not centriole recruitment that is affected by the mutation clusters. If this were the case it would be hard to discern an additional partial loss of centriole recruitment, as this loss would be masked by the PCM pool of dCPAP, especially under conditions of moderate overexpression of dCPAP. Importantly, however, our findings demonstrate that even very reduced amounts of centrosomal dCPAP can support robust centriole duplication as long as this protein can interact efficiently with dSTIL.

Our studies provide the first structural insight into the nature of the link between centrioles and human microcephaly. It is unclear why mutations in genes encoding key centriole or centrosome proteins can lead to such a specific neuro-developmental disorder in humans. It is widely assumed that some aspect of centriole/centrosome function must be particularly important in human neural progenitor cells, and that the failure of these cells to proliferate in an appropriate manner underlies the small brain size in affected individuals (*Megraw et al., 2011*). One possibility, based on the fact that

these neural progenitors seem to divide asymmetrically (*Siller and Doe, 2009*; *Megraw et al., 2011*), is that centrioles/centrosomes may play a particularly important role in properly orienting the spindle during asymmetric divisions, and division orientation could in turn be required for the maintenance of neuronal progenitors. This appears to be the case in flies, where mutations in dCPAP/DSas-4 and dSTIL/Ana2 lead to defects in the asymmetric division of the neural stem/progenitor cells (*Basto et al., 2006*; *Wang et al., 2011*). However, there are other possible explanations. Human neural progenitor cells form primary cilia, for example, and signalling through the cilium could be perturbed if centriole assembly is perturbed (*Han and Alvarez-Buylla, 2010*; *Megraw et al., 2011*). Moreover, several studies have linked centriole and centrosome malfunction to defects in DNA damage repair (DDR) pathways (*Megraw et al., 2011*), and mutations in MCPH genes can also lead to more severe pheno-types in humans that may be related to DDR pathway malfunction (*Al-Dosari et al., 2010*; *Kalay et al., 2011*; *Megraw et al., 2011*).

A previous analysis of the behaviour of various CPAP mutant proteins (modelled on MCPH mutations) in human cells revealed some surprising findings (*Kitagawa et al., 2011a*). The deletion of the TCP domain or the mutation of E1235 to Valine did not effect the localisation of CPAP to the centriole, although centriole duplication was compromised by both mutations. Moreover, overexpression of the E1235V mutant protein was able to promote centriole overgrowth to a greater extent than the WT protein, suggesting that it may have acquired some enhanced functionality. The structures we report here reveal that E1235 is one of the several residues involved in the binding interface with STIL, making an important sidechain–mainchain contact. This structural model explains how the E1235V mutation can compromise complex formation, and we have confirmed that this is the case with zebrafish proteins in vitro. Moreover, the equivalent mutation in flies leads to inefficient centriole assembly, but this process is not abolished. Taken together, our data strongly suggest that it is a partial failure in centriole assembly that is the primary cause of microcephaly in these patients. The challenge now is to understand how inefficient centriole assembly leads to microcephaly in humans.

## Materials and methods

### Recombinant protein expression and purification

*D. rerio* CPAP$^{937–1124}$ was cloned from *D. rerio* cDNA. Proteins were expressed in *Escherichia coli* BL21 (DE3) Rosetta as N-terminally His-tagged constructs, and purified via immobilised metal ion affinity chromatography (NiNTA; Qiagen, Hilden, Germany), proteolytic tag cleavage, followed (optionally) by size-exclusion chromatography and ion-exchange chromatography using standard methods. The selenomethionine derivative protein was expressed in selenomethionine supplemented M9 medium and purified in the same way. Purified constructs contained the sequence GPHM at the N-termini that stem from the cloning and protease cleavage sites.

*D. rerio* STIL$^{404–448}$ was cloned from IMAGE clone 7147918 and expressed in *E. coli* C41 BL21 (DE3), fused to two His-tagged lipoyl domains from *Bacillus stearothermophilus* dihydrolipoamide acetyltransferase at both the N- and C terminus. The peptide was purified via NiNTA chromatography, proteolytic cleavage of the His-lipoyl domains, and ion-exchange chromatography. The purified constructs contained a G (GG for *D. rerio* STIL$^{404–448}$ and its point-mutants) at their N-terminus and the sequence EFGENLYFQ (ENLYFQ for *D. rerio* STIL$^{408–428}$ and *D. rerio* STIL$^{411–428}$) at their C-terminus. These extra sequences stem from the cloning and protease cleavage sites.

Mutations of the *D. rerio* constructs were introduced into the expression vectors by site-directed mutagenesis.

Codon-optimised (GeneArt, Carlsbad, CA) *Drosophila* dSTIL$^{1–47}$ was genetically fused to the N-terminus of *Drosophila* dCPAP$^{700–901}$ via 3-way ligation. The fusion protein was expressed in *E. coli* B834 (DE3) as an N-terminally His-tagged fusion, and purified via NiNTA chromatography, proteolytic tag cleavage and size exclusion chromatography. The selenomethionine derivative protein was expressed using SelenoMethionine Medium (Molecular Dimensions, Newmarket, UK) and purified in the same way.

### Crystallisation

Native *D. rerio* CPAP$^{937–1124}$ was crystallised in sitting drops in 80 mM Tris pH 8.5, 160 mM MgCl$_2$, 20% PEG-4000, 18% Glycerol, 1 mM DTT at 19.5°C. The drops were set up using 1 µl of the protein solution and 0.5 µl of the reservoir solution. Crystals were mounted after 3 days and flash-frozen in liquid nitrogen.

**Table 2.** Native dataset analysis and refinement statistics

| | *D. rerio* CPAP[937–1124] WT | *D. rerio* CPAP[937–1124] E1021V | *D. rerio* CPAP[937–1124] + *D. rerio* STIL[408–428] complex | *D. melanogaster* dSTIL[1–47] – dCPAP[700–901] fusion complex |
|---|---|---|---|---|
| Beamline | Diamond I02 | MRC-LMB Cambridge UK | Diamond I04 | Diamond I04 |
| Space group | P21 | P21 | P21 | P1 |
| Wavelength (Å) | 0.9786 | 1.5418 | 0.9795 | 0.9795 |
| Monomers in the asymmetric unit | 1 | 1 | 2 | 3 |
| Unit cell dimensions (Å) | a = 52.34; b = 36.44; c = 56.44; α = 90.00; β = 117.31; γ = 90.00 | a = 52.12; b = 36.48; c = 56.46; α = 90.00; β = 117.47; γ = 90.00 | a = 60.25; b = 67.47; c = 61.65; α = 90.00; β = 113.92; γ = 90 | a = 58.64; b = 69.91; c = 69.98; α = 86.96; β = 88.64; γ = 67.69 |
| Resolution (Å) | 29.48–1.7 | 36.48–1.9 | 56.35–2.7/2.2 (anisotropy) | 64.60–2.57 |
| Completeness (overall/inner/outer shell) | 99.7/99.4/100 | 100/99.6/100 | 99.9/99.5/99.9 | 97.6/93.6/97.2 |
| Rmerge (overall/inner/outer shell) | 0.074/0.030/0.929 | 0.096/0.028/1.093 | 0.101/0.053/1.008 | 0.091/0.069/0.512 |
| Rpim (overall/inner/outer shell) | 0.029/0.012/0.369 | 0.039/0.012/0.456 | 0.050/0.027/0.505 | 0.061/0.035/0.449 |
| Mean I/σI (overall/inner/outer shell) | 14.6/39.9/2.0 | 13.8/43.4/1.8 | 7.6/19.0/1.4 | 7.6/16.3/1.7 |
| Multiplicity (overall/inner/outer shell) | 7.2/7.0/7.3 | 6.8/6.6/6.6 | 4.8/4.7/4.9 | 3.1/2.9/3.1 |
| Number of reflections | 19,941 | 14,349 | 21,892 | 31,911 |
| Number of atoms | 1595 | 1515 | 3176 | 3924 |
| Waters | 190 | 114 | 54 | 65 |
| Rwork/Rfree (% data used) | 19.9/24.4 (5.1%) | 20.9/26.7 (5.0%) | 23.4/27.7 (5.0%) | 24.5/26.3 (5.05%) |
| rmsd from ideal values: bond length/angles | 0.011/1.478 | 0.009/1.310 | 0.015/1.619 | 0.007/0.900 |
| Mean B value | 26.52 | 31.553 | 59.69 | 70.80 |
| Correlation coefficient Fo-Fc/Fo-Fc free | 0.961/0.942 | 0.955/0.926 | 0.954/0.933 | 0.854/0.836 |
| Molprobity Score | 0.97 (100th percentile) | 1.2 (99th percentile) | 1.70 (96th percentile) | 1.40 (100th percentile) |

*D. rerio* CPAP[937–1124] E1021V was crystallised in sitting drops in 80 mM Tris pH 8.5, 160 mM $MgCl_2$, 24% PEG-4000, 20% glycerol at 19.5°C. Crystals were mounted after 3 days and flash-frozen in liquid nitrogen.

SeMet *D. rerio* CPAP[937–1124] crystals were obtained using the sitting drop method with a reservoir solution of 80 mM Tris pH 8.5, 160 mM $MgCl_2$, 26% PEG-4000, 18% glycerol, 1 mM DTT at 19.5°C. Drops were set up using 1 µl protein solution and 1 µl of reservoir solution. Native CPAP[937–1124] crystals were used for streak-seeding into these drops and crystals allowed to grow for 7 days before mounting and flash-freezing them in liquid nitrogen.

Crystals of the complex of *D. rerio* CPAP[937–1124] and *D. rerio* STIL[408–428] were initially obtained using the LMB screening set-up with the Clear Strategy 2 pH 8.5 screen (MDL, Newmarket, UK). Crystals were used to streak seed into sitting drops consisting of 1 µl of a reservoir solution of 100 mM Tris pH 8.5, 200 mM CaAcetate, 17% PEG-2000 MME and 1 µl of protein/peptide mixture (0.25 µl protein + 0.75 µl peptide) at 19.5°C. Crystals were grown for 2 days before mounting them in 100 mM Tris pH 8.5, 200 mM CaAcetate, 17% PEG-2000 MME, 25% glycerol and flash-freezing them in liquid nitrogen.

**Table 3.** SeMet *D. rerio* CPAP[937–1124] dataset analysis and phasing statistics

| Beamline | ESRF ID 23–1 | | |
|---|---|---|---|
| Space group | P21 | | |
| Wavelength (Å) | 0.9791 (Peak) | 0.9794 (Inflection) | 0.9393 (Remote) |
| Unit cell dimensions (Å) | a = 52.39 b = 36.53 c = 56.34 α = 90.00 β = 117.28 γ = 90.00 | a = 52.59 b = 36.60 c = 56.48 α = 90.00 β = 117.24 γ = 90.00 | a = 52.49 b = 36.55 c = 56.38 α = 90.00 β = 117.26 γ = 90.00 |
| Resolution (Å) | 36.56–1.7 | 36.56–1.7 | 36.56–1.7 |
| Completeness (overall/inner/outer shell) | 100.0/99.7/100.0 | 100/99.2/100 | 100/99.7/100 |
| Rmerge (overall/inner/outer shell) | 0.09/0.048/1.296 | 0.127/0.047/2.840 | 0.092/0.046/1.370 |
| Rpim (overall/inner/outer shell) | 0.042/0.031/0.552 | 0.056/0.028/1.201 | 0.041/0.026/0.580 |
| Mean I/sd(I) (overall/inner/outer shell) | 10.7/26.0/1.5 | 8.9/26.4/0.7 | 10.8/27.4/1.4 |
| Multiplicity (overall/inner/outer shell) | 7.2/7.0/7.3 | 7.2/6.9/7.3 | 7.2/7.0/7.3 |
| Se sites found/expected | 5/7 | | |
| Overall FOM | 0.306 | | |

The protein concentrations of *D. rerio* CPAP[937–1124] used for crystallisations were measured by the Bradford assay with BSA as a standard and were 39.9 mg/ml (apo-CPAP[937–1124]), 46.2 mg/ml (CPAP[937–1124] E1021V), 30.6 mg/ml (SeMet CPAP[937–1124]) and 81 mg/ml CPAP[937–1124] (3.7 mM, CPAP/STIL complex). The concentration of STIL[408–428] (CPAP/STIL complex) was determined by amino acid analysis and was 11.8 mg/ml (3.8 mM).

Native *D. melanogaster* dSTIL[1–47]-dCPAP[700–901] was crystallised using the sitting drop approach, using the Morpheus screen (Molecular Dimensions). Crystals grew after approximately 3 weeks (*Table 5*, 'Native'). Crystals were mounted after approximately 4 weeks. SeMet *D. melanogaster*

**Table 4.** Characterisation of the CPAP:STIL interaction in vitro

| *Danio rerio* STIL[404–448] peptide in syringe | *Danio rerio* CPAP[937–1124] TCP domain in cell | Number of binding sites (N) | SD N | $K_D$ (µM) | SD $K_D$ (µM) | ΔH (kcal/mol) | SD (kcal/mol) | n (number of measurements) | Factor change in $K_D$ |
|---|---|---|---|---|---|---|---|---|---|
| WT | WT | 1.07 | 0.04 | 1.9 | 0.2 | −10.1 | 0.3 | 5 | 1 |
| WT | F978V | 0.70 | 0.09 | 37 | 10 | −23 | 5 | 4 | 20 |
| WT | T986V | 1.01 | 0.07 | 1.9 | 0.2 | −10.6 | 0.4 | 3 | 1 |
| WT | Y994V | 1.00 | 0.33 | 68 | 14 | −9.5 | 3.8 | 5 | 36 |
| WT | F1015V | 0.93 | 0.13 | 70 | 18 | −10.4 | 2.7 | 3 | 37 |
| WT | E1021V | 0.91 | 0.13 | 16 | 2 | −8.1 | 0.6 | 3 | 8 |
| WT | WT | 1.07 | 0.04 | 1.9 | 0.2 | −10.1 | 0.3 | 5 | 1 |
| P417A | WT | 1.06 | 0.02 | 37 | 1.3 | −11.5 | 0.2 | 3 | 20 |
| R418A | WT | 1.12 | 0.02 | 19 | 1 | −8.8 | 0.1 | 4 | 10 |
| P421A | WT | 1.16 | 0.03 | 17 | 0.3 | −9.8 | 0.2 | 4 | 9 |
| N422A | WT | 1.09 | 0.03 | 0.7 | 0.05 | −12.3 | 0.3 | 4 | 0.4 |
| P423A | WT | 1.16 | 0.05 | 4.6 | 0.3 | −10.9 | 0.4 | 4 | 2.4 |

Tables show the binding parameters between various *D. rerio* CPAP and STIL constructs obtained from ITC experiments. The measurements of the WT STIL[404–448]—WT CPAP[937–1124] interaction are identical to each other and identical to those shown in *Table 1* and are only presented again to allow easier comparison within each table. Fitting was performed with N as a variable. Constraining N to a fixed value of 1 during fitting produced $K_D$ values that were within the experimental error of those tabulated here. In control measurements on wild-type material and a selection of mutants of both CPAP and STIL, the experimental configuration was reversed with CPAP protein titrated into STIL peptide in the ITC cell. These experiments gave similar values for N, $K_D$ and ΔH to the standard configuration reported here.

**Table 5.** *D. melanogaster* dSTIL[1–47]-dCPAP[700–901] crystallisation conditions

| Crystal | Protein concentration (mg/ml) | Mother liquor | µl protein:µl Mother liquor | µl seed stock |
|---|---|---|---|---|
| Native | 6.18 | 100 mM MES/imidazole mix pH 6.5, 30 mM MgCl$_2$, 30 mM CaCl$_2$, 20% ethylene glycol, 10% PEG 8000 | 0.15:0.05 | – |
| Semet1 | 5.00 | 100 mM MES/imidazole mix pH 6.5, 20% ethylene glycol, 10% PEG8000, 0.2 M racemic glutamic acid, 0.2 M glycine, 0.2 M racemic serine, 0.2 M racemic alanine, 0.2 M racemic lysine HCl | 0.1:0.1 | – |
| Semet2 | 5.29 | 100 mM MES/imidazole mix pH 6.5, 14% ethylene glycol, 7% PEG8000, 30 mM NaNO$_3$, 30 mM NaPO$_4$, 30 mM NH$_4$SO$_4$ | 0.3:0.1 | 0.05 |
| Semet3 | 5.29 | 100 mM MES/imidazole mix pH 6.5, 14% ethylene glycol, 7% PEG8000, 30 mM NaNO$_3$, 30 mM NaPO$_4$, 30 mM NH$_4$SO$_4$ | 0.3:0.1 | 0.05 |
| Semet4 | 5.29 | 100 mM MES/imidazole mix pH 6.5, 16% ethylene glycol, 8% PEG8000, 30 mM NaNO$_3$, 30 mM NaPO$_4$, 30 mM NH$_4$SO$_4$ | 0.3:0.1 | 0.05 |

dSTIL[1–47]-dCPAP[700–901] was initially crystallised using the Morpheus screen (Molecular Dimensions). Crystals typically grew after 3–4 weeks. Some crystals were used for microseeding of further screens including an optimisation screen. Seed stock was generated using a Seed bead kit (Hampton, Aliso Viejo, CA). Details of crystallisation conditions are shown in *Table 5*.

**Table 6.** *D. melanogaster* dSTIL[1–47]-dCPAP[700–901] SeMet dataset analysis

| | Semet1-PEAK | SEMET1-LREM | Semet2-Peak | Semet3-PEAK | Semet3-INFL | Semet4-Peak |
|---|---|---|---|---|---|---|
| Beamline | Diamond IO4 | Diamond IO4 | Diamond IO3 | Diamond IO3 | Diamond IO3 | Diamond IO3 |
| Spacegroup | P1 | P1 | P1 | P1 | P1 | P1 |
| Wavelength | 0.9795 | 0.9999 | 0.9792 | 0.9791 | 0.9794 | 0.9791 |
| Unit cell dimensions (Å) | a = 59.31 b = 70.02 c = 70.01 α = 87.65 β = 89.24 γ = 67.37 | a = 59.14 b = 70.24 c = 70.13 α = 87.62 β = 89.12 γ = 67.35 | a = 58.47 b = 70.15 c = 69.99 α = 87.08 β = 88.41 γ = 67.60 | a = 58.56 b = 70.03 c = 70.14 α = 86.93 β = 88.39 γ = 68.09 | a = 58.72 b = 70.06 c = 70.28 α = 86.84 β = 88.47 γ = 68.36 | a = 59.01 b = 70.17 c = 70.15 α = 87.16 β = 88.64 γ = 67.58 |
| Resolution (Å) | 54.74–3.50 | 64.77–3.50 | 64.80–3.44 | 70.04–3.50 | 70.17–4.60 | 64.80–3.36 |
| Completeness (overall/inner/outer) | 98.1/93.8/98.3 | 98.4/98.0/98.2 | 97.8/91.6/93.7 | 98.3/95.3/97.6 | 97.8/79.1/89.9 | 97.6/91.0/97.4 |
| Rmerge (overall/inner/outer) | 0.093/0.055/0.118 | 0.086/0.041/0.238 | 0.17/0.076/0.518 | 0.152/0.086/0.336 | 0.116/0.039/0.189 | 0.125/0.037/0.433 |
| Rpim (overall/inner/outer) | 0.071/0.047/0.136 | 0.062/0.029/0.172 | 0.078/0.040/0.229 | 0.075/0.043/0.184 | 0.087/0.034/0.141 | 0.100/0.038/0.323 |
| I/σI (overall/inner/outer) | 9.1/16.9/5.6 | 10.9/24.7/4.9 | 7.3/18.8/3.5 | 9.1/27.2/3.6 | 6.0/20.8/4.6 | 6.9/22.5/2.7 |
| Multiplicity (overall/inner/outer) | 3.9/3.8/3.9 | 3.9/3.8/3.8 | 7.0/7.0/7.1 | 6.0/6.7/5.3 | 3.5/3.6/3.5 | 3.5/3.4/3.6 |
| No. unique reflections | 12,832 | 12,884 | 13,315 | 12,797 | 5641 | 14,461 |

**Table 7.** Quantification of centriole/centrosome numbers in *dCPAP* or *dSTIL* mutant larval brain cells expressing the indicated WT or mutant constructs

| Genotype | Number of brains | Total number of cells | Cells with centrosome number (%) | | | |
|---|---|---|---|---|---|---|
| | | | 0 | 1 | 2 | 3 |
| WT | 12 | 944 | 2.1 | 2.6 | 95.2 | 0.0 |
| *dCPAP* | 8 | 661 | 95.2 | 4.2 | 0.6 | 0.0 |
| dCPAP_WT-GFP | 9 | 715 | 2.8 | 6.3 | 90.5 | 0.4 |
| dCPAP_ΔC-GFP | 13 | 1147 | 95.1 | 3.8 | 1.0 | 0.0 |
| dCPAP_MC1-GFP | 11 | 968 | 64.9 | 30.1 | 4.8 | 0.3 |
| dCPAP_MC2-GFP | 17 | 1053 | 43.5 | 42.2 | 14.1 | 0.3 |
| dCPAP_MC3-GFP | 16 | 1870 | 4.5 | 13.3 | 81.9 | 0.3 |
| dCPAP_MC1-3-GFP | 11 | 888 | 90.1 | 8.0 | 1.8 | 0.1 |
| dCPAP_E792V-GFP | 9 | 1015 | 13.7 | 31.1 | 54.8 | 0.4 |
| *dSTIL*[169] | 9 | 846 | 98.1 | 1.9 | 0.0 | 0.0 |
| *dSTIL*[719] | 9 | 980 | 99.2 | 0.8 | 0.0 | 0.0 |
| dSTIL_WT-GFP | 6 | 424 | 5.7 | 9.0 | 85.1 | 0.2 |
| dSTIL_ΔN-GFP | 13 | 884 | 88.1 | 9.8 | 1.9 | 0.1 |
| dSTIL_P11A-GFP | 9 | 1008 | 11.0 | 41.0 | 48.0 | 0.0 |
| dSTIL_R12A-GFP | 9 | 709 | 7.0 | 31.0 | 62.0 | 0.0 |
| dSTIL_P11AR12A-GFP | 9 | 727 | 41.0 | 43.0 | 16.0 | 0.0 |

## Data collection and processing

Native data were collected as described in *Table 2*. All *D. rerio* datasets were integrated and scaled using MOSFLM (*Leslie and Powell, 2007*) and Scala (*Evans, 2006*) respectively. The *D. rerio* CPAP[937–1124] structure was solved by MAD in CRANK (*Ness et al., 2004*; *Cowtan, 2006*), resulting in clear electron density into which an initial model was built using ArpWarp (*Langer et al., 2008*). Phenix.refine (*Afonine et al., 2005*) and REFMAC (*Murshudov et al., 2011*) were used to refine the model against the native dataset with manual building done in Coot (*Emsley and Cowtan, 2004*). *D. rerio* CPAP[937–1124] E1021V was solved by molecular replacement in Phaser (*McCoy et al., 2007*) using a poly-alanine model derived from the WT model. The model was further built and refined as described for the WT structure. The complex of *D. rerio* CPAP[937–1124] and *D. rerio* STIL[408–428] was solved by molecular replacement using Phaser (*McCoy et al., 2007*) with a distorted model of the *D. rerio* CPAP[937–1124] WT apo-structure. Refinement yielded clear density for the residues of STIL shown here. The model was further built and refined as described for the other *D. rerio* structures.

*D. melanogaster* dSTIL[1–47]-dCPAP[700–901] data was scaled using Xia2 (*Winter, 2010*). Phasing was carried out using all SeMet datasets (*Table 6*) in autoSHARP (*Vonrhein et al., 2007*), using SHELXC/D (*Sheldrick, 2008*) for heavy atom finding, SHARP for site refinement/phasing and SOLOMON (*Abrahams and Leslie, 1996*) for density modification. This resulted in an experimental density map within which a CHAINSAW (*Stein, 2008*) model based on the *D. rerio* complex structure could be manually placed, using heavy atom sites as a guide. Experimental density corresponding to the dSTIL peptide could be easily seen. Further refinement cycles allowed the remaining copies of the monomer to be placed and trimmed. Refinement and model building were carried out in autoBUSTER (*Bricogne et al., 2011*) and Coot (*Emsley and Cowtan, 2004*) respectively.

## Isothermal calorimetry (ITC) measurements

All ITC measurements were performed using an auto-iTC 200 instrument (GE Healthcare, Little Chalfont, UK) in 50 mM HEPES pH 7.5, 100 mM NaCl at 25°C. Samples were stored by the instrument

in 96-well microtiter plates at 5°C prior to loading and performing the titrations. Standard experiments used 19 × 2 µl injections of STIL peptide into CPAP protein preceded by a single 0.5 µl pre-injection. Heat from the pre-injection was not used during fitting. Data were analysed manually in the Origin software package provided by the manufacturer and fit to a single set of binding sites model. All measurements were corrected using control ITC experiments in which the peptide studied was injected into buffer only. The small endothermic heats of injection in these experiments were fitted to a linear function that was subsequently subtracted from the equivalent integrated heats of the peptide–protein binding experiment before fitting. The concentration of CPAP in the cell was typically 40 µM but varied maximally between 20 and 100 µM. The concentration of STIL used in the syringe was typically 700 µM but varied maximally between 600 and 2600 µM depending on the affinity of the peptide interaction being studied.

## In vivo analysis in *Drosophila*

### Fly stocks and transgenic constructs

The following mutant alleles and stocks were used in this study: $ana2^{169}$ (here called $dSTIL^{169}$), $ana2^{719}$ (here called $dSTIL^{719}$) (*Wang et al., 2011*), $sas-4^{s2214}$ (here called $dCPAP$) (*Basto et al., 2006*), pUbq-dCPAP_WT-GFP, pUbq-dCPAP_$\Delta C_{1–724}$-GFP, pUbq-dCPAP_MC1-GFP, pUbq-dCPAP_MC2-GFP, pUbq-dCPAP_MC3-GFP, pUbq-dCPAP_MC1-3-GFP, pUbq-dCPAP_E792V-GFP, pUbq-dSTIL_WT-GFP, pUbq-dSTIL_$\Delta N_{46–420}$-GFP, pUbq-dSTIL_P11A-GFP, pUbq-dSTIL_R12A-GFP, pUbq-dSTIL_P11AR12A-GFP. Transgenic lines contain GFP fused to the C-terminus of dCPAP and dSTIL, respectively, and are expressed from the Ubiquitin promoter, which drives moderate expression in all cell types (*Lee et al., 1988*). Flies were kept at 25°C, OregonR served as wild-type control.

GFP-tagged full length versions of dCPAP and dSTIL used in this study were made by cloning the full length dCPAP cDNA and the dSTIL cDNA into the pUbq-GFP(C-terminus) destination vector using the Gateway System (Life Technologies, Carlsbad, CA). PCR with dCPAP_WT and dSTIL_WT as template was used to make dCPAP_$\Delta C_{1–724}$ and dSTIL_$\Delta N_{46–420}$, respectively. Single point mutations and mutation clusters were introduced into full length dCPAP and dSTIL using site-directed mutagenesis (QuickChange II XL/Quick Change Lightening Multi Site-Directed Mutagenesis Kits, Agilent Technologies, Santa Clara, CA). Constructs were injected either by Genetic Services Inc. (Cambridge, MA) or Cambridge DNA Injection Service (Cambridge, UK).

### Rescue experiments

All constructs were tested for their ability to rescue the uncoordinated phenotype, which is a feature of flies lacking centrioles (*Basto et al., 2006*). For that purpose, the different versions of dCPAP-GFP and dSTIL-GFP were either crossed into the $dCPAP$ or $dSTIL^{169}$/$dSTIL2^{719}$ mutant background, and desired pupae were collected from vials and transferred to filter paper (Whatman, Maidstone, UK) for analysis.

### Immunohistochemistry on third instar larval brains and centrosome quantification

Brains were dissected, squashed, and stained as previously described (*Stevens et al., 2009*). The following antibodies were used to stain centrosomes in third instar larval brain cells: sheep anti-Centrosomin (Cnn, directed against the N-terminus, 1:1000, [*Lucas and Raff, 2007*] but raised in sheep), guinea pig anti-Asterless (Asl, 1:500, [*Conduit et al., 2010*] but raised in guinea pig). Secondary antibodies conjugated to either Alexa Fluor 488 or Alexa Fluor 568 (Life Technologies) were used 1:1000. Hoechst33258 (Life Technologies) was used to visualise DNA (1:5000). Centrosomes were counted on a Zeiss Axioskop 2 microscope (Zeiss, Oberkochen, Germany). Only brain cells in metaphase were scored that did stain for Asl and Cnn. DNA morphology was used to identify cells at the desired stage of the cell cycle. Furthermore, the assessment of centriole loss was performed blind. Microsoft Excel was used to analyse the data. Images were acquired in Metamorph (molecular devices) using a CoolSNAP HQ camera (Photometrics, Tucson, AZ) and processed using ImageJ/Fiji (www.fiji.sc/Fiji, [*Schindelin et al., 2012*]), Gimp (www.gimp.org/) and Inkscape (www.inkscape.org/) for figure assembly.

### Western blot analysis

The following primary and secondary antibodies were used: Mouse anti-GFP (1:500, Roche, Basel, Switzerland), mouse anti-actin (1:1000, SIGMA, St. Louis, MO), rabbit anti-dCPAP (1:500)

(*Basto et al., 2006*), anti-mouse HRP (1:3000, GE Healthcare) and anti-rabbit HRP (1:3000, GE Healthcare).

## Live imaging of embryos
Embryos expressing the different GFP-tagged versions of dCPAP and dSTIL were dechorionated manually and mounted in a Glass Bottom Microwell Dish (MatTek, Ashland, MA) using heptane glue. Embryos were covered with voltalef oil and followed by time-lapse spinning disc microscopy on a Perkin Elmer spinning disc microscope (Perkin Elmer, Waltham, MA). Images were acquired with a charge-coupled Orca ER device camera (Hamamatsu Photonics, Hamamatsu, Japan) using UltraView ERS (Perkin Elmer) and processed and analysed in Velocity (Perkin Elmer).

## *C. elegans* experiments
*C. elegans* strains carrying single-copy *sas-4* transgenes were generated using MosSCI (*Frøkjær-Jensen et al., 2008*). To render the transgenes RNAi-resistant, a 500 bp region at the 5' end of the *sas-4* genomic sequence was re-encoded. The engineered *sas-4* sequence was cloned into pCFJ151 with the promoter and 3' UTR from *sas-6*, as well as a C-terminal GFP tag. pCFJ151 contains homology arms that direct transposase-mediated insertion of intervening sequence into the ttTi5606 Mos1 site on Chromosome II. Transgene integration was confirmed by PCR of regions spanning each side of the insertion. The genotypes of the strains used are: unc-119(ed9)III; ltSi85[pOD1550; Psas-6::SAS-4 reencoded::GFP; cb-unc-119(+)]II for WT SAS-4; and unc-119(ed9)III; ltSi177[pOD1551; Psas-6::SAS-4(1-556) reencoded::GFP; cb-unc-119(+)]II for SAS-4$^{\Delta TCP}$.

Double-stranded *sas-4* RNA was generated as described (*Oegema et al., 2001*) using DNA templates prepared by PCR. For experiments to quantify monopolar spindle formation, L4 hermaphrodites were injected with dsRNA and incubated at 20°C for 40 hr prior to dissection for imaging. For lethality assays, worms were maintained at 20°C. L4 hermaphrodites were injected with dsRNA and singled 24 hr post-injection. Adult worms were removed from the plates 48 hr post-injection, and hatched larvae and unhatched embryos were counted 24 hr later.

For light microscopy to identify monopolar or bipolar second division cells, images were acquired using an inverted Zeiss Axio Observer Z1 system with a Yokogawa spinning-disk confocal head (CSU-X1), a 63X 1.4 NA Plan Apochromat objective, and a QuantEM:512SC EMCCD camera (Photometrics). Adult worms were dissected in M9 buffer, and embryos were mounted onto 2% agarose pads for imaging. 11 × 1 µm z-stacks were collected in the GFP channel (100 ms, 20% power, no binning), along with one central DIC section.

## SAS-4/SAS-5 pull-down experiments
SAS-4 constructs were cloned into a pET21a vector for in vitro transcription/translation. Proteins were expressed using the T7 TNT Quick Coupled Transcription/Translation System (Promega, Fitchburg, WI) with $^{35}$S-Met labelling.

SAS-5 fragments were cloned into a pRSET-A vector with a C-terminal 6xHis tag. Proteins were expressed in *E. coli* Rosetta2(DE3) cells and purified on Ni-NTA agarose (Qiagen) using standard protocols. For pull-down experiments, proteins were dialysed into 25 mM HEPES, 100 mM NaCl, 20 mM imidazole, 1 mM DTT, 10% sucrose, 0.02% Tween-20, pH 7.4.

SAS-5 fragments were pre-incubated with 20 µl Ni-NTA beads for 45 min at 4°C. 10 µl of the SAS-4 IVTT product was added to the beads with 190 µl buffer and incubated at 4°C for 30 min. The beads were washed with 3 × 200 µl buffer and resuspended in 100 µl SDS-PAGE sample buffer. Samples were run on 10% SDS-PAGE gels and either stained with Coomassie or dried and exposed to a phosphor screen overnight. Phosphor screens were analysed on a Personal Molecular Imager System (Bio-Rad, Hercules, CA).

# Acknowledgements
For beamline support, MVB would like to acknowledge Dr Pierpaolo Romano (I04) and Dr Thomas Sorensen (I02) at Diamond Light Source, Oxford, UK, and Dr Alexander Popov (ID23-1) and Petra Pernot (ID14–3) at the European Synchrotron Radiation Facility (ESRF), Grenoble, France. We also would like to acknowledge Meindert Lamers (MRC-LMB, Cambridge, UK) for his help with SAXS data processing. SML and MAC would like to acknowledge their beamline supports at beamlines I03 and I04 at Diamond Light Source, Oxford, UK. The structures presented in this work have been deposited under PDB codes 4bxp, 4bxq, 4bxr, and 4by2.

# Additional information

## Funding

| Funder | Grant reference number | Author |
|---|---|---|
| UK Medical Research Council | MC_UP_1201/3 | Mark van Breugel |
| Cancer Research UK Program Grant | 10530 | Jordan W Raff |
| Wellcome Trust | 083599/Z/07/Z | Susan M Lea |
| UK Medical Research Council | G0900888 | Steven Johnson |
| National Institutes of Health | R01-GM074207 | Karen Oegema |
| Ludwig Institute for Cancer Research | | Karen Oegema |
| Biotechnology and Biological Sciences Research Council UK Stipend | | Matthew A Cottee |

The funders had no role in study design, data collection and interpretation, or the decision to submit the work for publication.

## Author contributions

MAC, SML, SJ, Crystallised and solved the *Drosophila* dCPAP–STIL complex; MAC, SJ, Biophysical characterisations of the *Drosophila* proteins that were essential for the crystallisation work; MAC, NM, Cloned and generated transgenic fly lines; NM, JWR, Contributed the *Drosophila* in vivo work; YLW, KO, Contributed the *C. elegans* work; CMJ, MvB, Performed the ITC experiments with zebrafish CPAP and STIL; AA, Contributed the sequence alignments and conservation analysis; MvB, Crystallised and solved the zebrafish structures; MAC, NM, YLW, CMJ, SJ, AA, KO, SML, JWR, MvB, Contributed to the writing of the manuscript

# Additional files

## Major datasets

The following datasets were generated:

| Author(s) | Year | Dataset title | Dataset ID and/or URL | Database, license, and accessibility information |
|---|---|---|---|---|
| van Breugel M | 2013 | Structure of the wild-type TCP10 domain of *Danio rerio* CPAP | http://www.rcsb.org/pdb/search/structidSearch.do?structureId=4bxp | Publicly available at the RCSB Protein Data Bank (http://www.rcsb.org/). |
| van Breugel M | 2013 | Structure of the E1021V mutant of the TCP10 domain of Danio rerio CPAP | http://www.rcsb.org/pdb/search/structidSearch.do?structureId=4bxq | Publicly available at the RCSB Protein Data Bank (http://www.rcsb.org/). |
| van Breugel M | 2013 | Structure of the wild-type TCP10 domain of *Danio rerio* CPAP in complex with a peptide of *Danio rerio* STIL | http://www.rcsb.org/pdb/search/structidSearch.do?structureId=4bxr | Publicly available at the RCSB Protein Data Bank (http://www.rcsb.org/). |
| Cottee MA, Lea SM | 2013 | SAS-4 (dCPAP) TCP domain in complex with a Proline Rich Motif of Ana2 (dSTIL) of *Drosophila melanogaster* | http://www.rcsb.org/pdb/search/structidSearch.do?structureId=4by2 | Publicly available at the RCSB Protein Data Bank (http://www.rcsb.org/). |

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
