## [Decision Letter]

Thank you for sending your work entitled “Crystal structures of the CPAP/STIL complex reveal its role in centriole assembly and human microcephaly” for consideration at *eLife*. Your article has been favorably reviewed by an editor and a reviewer. The Senior editor and the other reviewer discussed their comments before we reached this decision, and the Senior editor has assembled the following comments to help you prepare a revised submission.

This manuscript by Cottee et al. offers the first structural insights into the link between centrioles and microcephaly, based on detailed experiments ranging from structural biology and bioinformatics to cell biology and in vivo studies. The cornerstone is the identification of the molecular units that interact in centriole assembly. Evidence is presented for the interaction and it is illuminated by atomic structures.

In this paper the authors provide crystal structures for a central part of the CPAP/STIL protein complex from *Drosophila* and zebrafish. The CPAP/STIL complex is involved in centriole formation. The structures reveal a 20-strand beta sheet that forms a binding pocket for an extended polyproline motif. From a structural perspective, this is most interesting, since the STIL-interacting domain of CPAP (the TCP domain) forms a single beta sheet that is reminiscent of the structure of amyloid fibrils. The distinction is that amyloid fibrils are formed by strands from multiple protein molecules, whereas here a single protein forms the sheet. Also, in amyloid fibers the sheets pack against each other whereas the CPAP TCP domain forms a single sheet. This structure may well be unprecedented, and the finding is important because it extends our understanding of protein architecture.

The TCP domain is found to form a novel proline-recognition domain. The authors identified a ∼40 amino acid region in STIL that is proline rich and conserved. They dissected this region to find that a peptide segment corresponding to the entire region binds with 2µM affinity to CPAP, but that a shorter segment (411–428) binds with only slightly diminished affinity. This shorter segment was crystallised with the TCP domain, using the zebrafish sequences, and also on its own. Remarkably, the single beta sheet formed by the TCP domain lacks a hydrophobic core, and is stabilised principally by hydrogen bonding. Sequence repeats have been identified previously in the domain and these are shown to form the beta-hairpin repeats in the structure. There are no extensive crystal packing interfaces, and so the structure suggests that the functional unit of TCP is a monomer. This is verified by biophysical data.

The STIL peptide adopts a PP-II helix conformation and interacts with the TCP domain via conventional interactions, including the packing of proline side-chains from the peptide into hydrophobic surfaces on the TCP domain. Additional conservation in the TCP domain, beyond that required for the interaction with the short crystallised peptide, suggests that the complete interface is more extensive.

The importance of the crystallographic interface between the TCP domain and the STIL peptide has been verified by mutagenesis and by determining the structure of the equivalent complex from *Drosophila*, revealing strong structural conservation despite weak sequence similarity. There is extensive analysis of the implications of the structure for centriole formation, using cell-based assays in the *Drosophila* system.

The novelty of the structural fold of the TCP domain and the new mechanism revealed for recognition of proline-rich peptides revealed by the structures of the complexes makes this paper suitable for *eLife*. The paper represents a significant advance, and it is well validated and well described.

Minor comments:

1) In one place the authors call the interaction with the peptide 'unique' and in another 'highly unusual.' It would strengthen the paper to give quantitative evidence for the lack of similarity of this structural element to previously determined structures, and for a brief discussion of any similar structures. At the very least, the authors may wish to give somewhat more prominence to their very unusual structural finding. It is quite rare these days to determine a structure of a soluble protein and really find something new.

2) The paper is very clearly written and can be accepted with very minor changes. One suggestion regarding clarity is that the paper will be of interest to structural biologists who are not familiar with the cell-based assay systems, and so these should be explained better. To start with, the “unc” phenotype is mentioned in the Results with no explanation. Presumably, this is the “uncoordinated phenotype” mentioned in the earlier paragraph, but this should be explained. The use of this phenotype in the cell-based assays needs more explanation. Presumably, a general behavioral phenotype (“uncoordinated”) is being mapped into specific features of the embryonic cells?

---

## [Author Response]

*1) In one place the authors call the interaction with the peptide 'unique' and in another 'highly unusual.' It would strengthen the paper to give quantitative evidence for the lack of similarity of this structural element to previously determined structures, and for a brief discussion of any similar structures. At the very least, the authors may wish to give somewhat more prominence to their very unusual structural finding. It is quite rare these days to determine a structure of a soluble protein and really find something new*.

We expanded the structural description of the TCP domain and included an additional figure supplement with information on its sequence repeats. We also commented on the similarity to other protein structures and added a figure supplement showing a side-by-side comparison of the TCP domain and the closest structural match in the PDB. It is noteworthy to mention that this structure is artificially engineered and is capped on both ends by globular domains that stabilise it.

The TCP domain folds into an extended β-sheet meander. In general, β-meanders are frequently found in the PDB, e.g., in β-barrels , β-propellers or some α+β proteins where often α-helices pack against the meander sheet. What makes the TCP domain structure unique is that it consists solely of a freestanding meander β-sheet that is on both sides exposed to the solvent and entirely lacks a defined hydrophobic core. We also demonstrate that this domain is stable on its own and it is folded and predominantly monomeric in solution. Currently, there are only three natural proteins, structurally characterised and deposited in the PDB, that contain regions forming an elongated single meander β-sheet. These are: outer surface protein A (1osp), histone methyltransferase set7/9 (1h3i) and bacteriophage gp138 (PDB code 3pqh). In none of these, however, the meander β-sheet exists on its own. The 'nonglobular' β-sheet in each of these structures is flanked by other globular domains and in two of them the region of the β-sheet that is exposed to the solvent is relatively small (three strands in OspA and set7/9). In the bacteriophage gp138, one of the β-sheet faces is hydrophobic and buried in the trimer interface.

As to quantitative evidence for the lack of similarity to other structures in the PDB, we decided not to include it as it would not be very meaningful in our opinion. We could have included the RMSDs of comparisons to other β-meander containing proteins. However, what would these numbers really mean, if the overall protein architecture and the context in which the meander is found are so different? We could have used the Z-scores of the DALI server (or the equivalent scores of similar comparison servers). However, these scores suffer from similar shortcomings. Furthermore, these servers are well known to miss structural similarities and it is not clear to us therefore what the absence of close hits would signify. We would like to point out that one of the co-authors of this paper (AA) is a developer of the SCOP (Structural Classification of Proteins) database and therefore has a good overview of the available protein structural repertoire.

*2) The paper is very clearly written and can be accepted with very minor changes. One suggestion regarding clarity is that the paper will be of interest to structural biologists who are not familiar with the cell-based assay systems, and so these should be explained better. To start with, the “unc” phenotype is mentioned in the Results with no explanation. Presumably, this is the “uncoordinated phenotype” mentioned in the earlier paragraph, but this should be explained. The use of this phenotype in the cell-based assays needs more explanation. Presumably, a general behavioral phenotype (“uncoordinated”) is being mapped into specific features of the embryonic cells*?

We clarified the corresponding sections in detail to make the manuscript clearer and easier to follow for non-*Drosophila* experts. Please also note that we include in our manuscript (Figure 3) three more *Drosophila* STIL mutants (P11A, R12A and P11AR12A) that we completed analyzing while our manuscript was under review. The corresponding phenotypes of these mutants further strengthen our argument and strongly confirm our structural model.